# The COVID-19 Pandemic Impacted Maternal Mental Health Differently Depending on Pregnancy Status and Trimester of Gestation

**DOI:** 10.3390/ijerph19052926

**Published:** 2022-03-02

**Authors:** Anick Bérard, Jessica Gorgui, Vanina Tchuente, Anaïs Lacasse, Yessica-Haydee Gomez, Sylvana Côté, Suzanne King, Flory Muanda, Yves Mufike, Isabelle Boucoiran, Anne Monique Nuyt, Caroline Quach, Ema Ferreira, Padma Kaul, Brandace Winquist, Kieran J. O’Donnell, Sherif Eltonsy, Dan Chateau, Jin-Ping Zhao, Gillian Hanley, Tim Oberlander, Behrouz Kassai, Sabine Mainbourg, Sasha Bernatsky, Évelyne Vinet, Annie Brodeur-Doucet, Jackie Demers, Philippe Richebé, Valerie Zaphiratos

**Affiliations:** 1Faculty of Pharmacy, University of Montreal, Montreal, QC H3T 1J4, Canada; jessica.gorgui@umontreal.ca (J.G.); ema.ferreira@umontreal.ca (E.F.); 2Research Center, CHU Sainte-Justine, Montreal, QC H3T 1C5, Canada; vanina.tchuente.hsj@ssss.gouv.qc.ca (V.T.); yessica-haydee.gomez.hsj@ssss.gouv.qc.ca (Y.-H.G.); sylvana.cote.1@umontreal.ca (S.C.); isabelle.boucoiran@umontreal.ca (I.B.); anne.monique.nuyt.med@ssss.gouv.qc.ca (A.M.N.); c.quach@umontreal.ca (C.Q.); jinping.zhao@mail.mcgill.ca (J.-P.Z.); 3Faculty of Medicine, Université Claude Bernard Lyon 1, 69003 Lyon, France; sabine.mainbourg@chu-lyon.fr; 4Health Sciences Department, Université du Québec en Abitibi-Témiscamingue, Rouyn-Noranda, QC J9X 5E4, Canada; anais.lacasse@uqat.ca; 5Faculty of Medicine, School of Public Health, University of Montreal, Montreal, QC H3T 1J4, Canada; 6Faculty of Medicine, McGill University, Montreal, QC H3G 2M1, Canada; suzanne.king@mcgill.ca; 7Department of Epidemiology & Biostatistics, Western University, London, ON N6A 5W9, Canada; flory.muanda-tsobo@lhsc.on.ca; 8ICES Western, Western University, London, ON N6A 5W9, Canada; 9Department of Family Medicine, Protestant University in Congo, Kinshasa II, Democratic Republic of the Congo; dr.mufike@hotmail.fr; 10School of Public Health, Department of Obstetrics and Gynecology, University of Montreal, Montreal, QC H3N 1X9, Canada; 11Department of Pediatrics, CHU Sainte-Justine, University of Montreal, Montreal, QC H3T 1C5, Canada; 12Department of Microbiology, Infectious Diseases and Immunology, University of Montreal, Montreal, QC H3T 1J4, Canada; 13Pharmacy Department, CHU Sainte-Justine, Montreal, QC H3T 1C5, Canada; 14Department of Medicine, 4-120 Katz Group Centre for Pharmacy and Health Research, University of Alberta, Edmonton, AL T6G 2R7, Canada; pkaul@ualberta.ca; 15College of Medicine, Department of Community Health and Epidemiology, University of Saskatcheawan, Saskatoon, SK S7N 5E5, Canada; brandy.winquist@usask.ca; 16Yale Child Study Center, Department of OB/GYN and Reproductive Sciences, Yale School of Medicine, New Haven, CT 06510, USA; kieran.odonnell@yale.edu; 17Douglas Research Center, Department of Psychiatry, McGill University, Montreal, QC H4H 1R3, Canada; 18College of Pharmacy, Rady Faculty of Health Sciences, University of Manitoba, Winnipeg, MB R3E 0W2, Canada; sherif.eltonsy@umanitoba.ca; 19Manitoba Center for Health Policy, Winnipeg, MB R3E 3P5, Canada; dan.chateau@anu.edu.au; 20Department of Obstetrics & Gynaecology, University of British Columbia, Vancouver, BC V6T 1Z4, Canada; gillian.hanley@vch.ca; 21Department of Pediatrics, School of Population and Public Health, University of BC, Vancouver, BC V6T 1Z4, Canada; toberlander@bcchr.ca; 22Department of Clinical Epidemiology, UMR 5558 CNRS, Clinical Investigation Centre, Inserm-Hospices Civils de Lyon, Claude Bernard University Lyon 1, 69000 Lyon, France; behrouz.kassai-koupai@chu-lyon.fr; 23Divisions of Clinical Epidemiology and Rheumatology, McGill University Health Centre, Montreal, QC H3A 0G4, Canada; sasha.bernatsky@mcgill.ca (S.B.); evelyne.vinet@mcgill.ca (É.V.); 24Dispensaire Diététique de Montréal/Montreal Diet Dispensary, Montreal, QC H3H 1J3, Canada; abdoucet@dispensaire.ca (A.B.-D.); jdemers@dispensaire.ca (J.D.); 25Department of Anesthesiology and Pain Medicine, CIUSSS de l’Est de l’Ile de Montreal, Maisonneuve-Rosemont Hospital, University of Montreal, Montreal, QC H1T 2M4, Canada; philippe.richebe@umontreal.ca; 26Research Center, Maisonneuve-Rosemont Hospital, CIUSSS de l’Est de l’Ile de Montreal, Montreal, QC H1T 2M4, Canada; valerie@zaphiratos.ca

**Keywords:** COVID-19 pandemic, maternal mental health, pregnancy and delivery, trimester of pregnancy, Edinburgh Perinatal Depression Scale (EPDS), Generalized Anxiety Disorders (GAD-7), stress

## Abstract

Introduction: We aimed to measure the impact of the COVID-19 pandemic on maternal mental health, stratifying on pregnancy status, trimester of gestation, and pandemic period/wave. Methods: Pregnant persons and persons who delivered in Canada during the pandemic, >18 years, were recruited, and data were collected using a web-based strategy. The current analysis includes data on persons enrolled between 06/2020–08/2021. Maternal sociodemographic indicators, mental health measures (Edinburgh Perinatal Depression Scale (EPDS), Generalized Anxiety Disorders (GAD-7), stress) were self-reported. Maternal mental health in pregnant women (stratified by trimester, and pandemic period/wave at recruitment) was compared with the mental health of women who had delivered; determinants of severe depression were identified with multivariate logistic regression models. Results: 2574 persons were pregnant and 626 had already delivered at recruitment. Participants who had delivered had significantly higher mean depressive symptom scores compared to those pregnant at recruitment (9.1 (SD, 5.7) vs. 8.4 (SD, 5.3), *p* = 0.009). Maternal anxiety (aOR 1.51; 95%CI 1.44–1.59) and stress (aOR 1.35; 95%CI 1.24–1.48) were the most significant predictors of severe maternal depression (EDPS ˃ 13) in pregnancy. Conclusion: The COVID-19 pandemic had a significant impact on maternal depression during pregnancy and in the post-partum period. Given that gestational depression/anxiety/stress has been associated with preterm birth and childhood cognitive problems, it is essential to continue following women/children, and develop strategies to reduce COVID-19′s longer-term impact.

## 1. Introduction

COVID-19-related public health measures and pandemic-related stressors have contributed to emotional distress, particularly depression and anxiety [1,2]. Interventions such as indefinite confinement and conflicting public health messaging intensified distress, especially among vulnerable populations including pregnant persons [3]. After the severe acute respiratory syndrome epidemic in 2003, non-pregnant persons experienced anxiety, stress, and depression for these reasons [4]. In addition, since the start of the COVID-19 pandemic, perinatal care strategies have been revised to decrease in-person visits, which may increase stress and anxiety among pregnant persons as well as hesitation to attend the few necessary clinic visits due to fear of infection, further increasing risk for adverse obstetrical and post-partum outcomes [5].

Prenatal exposure to crises can have lasting effects on the cognitive and behavioral health of pregnant persons and their newborns [6,7,8,9,10]. In Project Ice Storm (1 million Quebec/Ontario residents without electricity, 6 January to 6 February 1998), studies [7,11,12] showed that greater maternal depression and anxiety from the crisis during pregnancy significantly predicted poor outcomes in children, such as lower IQ and altered brain development in children exposed to stress *in utero*. Furthermore, maternal stress, depression, and anxiety during pregnancy have been independently shown to be associated with low birth weight [13], prematurity [1], and post-partum depression [14]. At present, studies on COVID-19 and pregnancy focus primarily on obstetrical risk to the mother and fetus with little data on pregnant persons’ mental health. Current findings (i) vary depending on the place of residence, (ii) include small sample sizes, (iii) are from hospital samples that are likely not representative of the population, or (iv) do not consistently use standardized measures of mental health. [5,15,16] We know that 37% of pregnant persons in Alberta reported severe depressive symptoms, and 57% severe anxiety symptoms during pregnancy soon after the first COVID-19 cases were identified in Canada [15]. However, no large-scale Canadian study has evaluated the impact of the pandemic on mental health during and after pregnancy, across the first three waves of the COVID-19 pandemic. 

Given the reported impact of the pandemic on mental health in the general non-pregnant population [17] and that specific fetal developments occur at different time-points during pregnancy (the first trimester relates to organogenesis, 2nd and 3rd trimesters to prematurity, low birth weight, and cognitive function in children), we aimed to quantify the prevalence of maternal depression, anxiety and stress stratified on the status of pregnancy (pregnant vs. delivered) and trimester of gestation as well as pandemic period/wave, and identify predictors of severe gestational maternal depression during the COVID-19 pandemic.

## 2. Materials and Methods 

### 2.1. Study Design 

The recruitment started on 26 June 2020 and is ongoing. The present analysis includes cross-sectional data on Canadians enrolled between 26 June–17August 2021. Participants could enter the study at various moments: cohort 1—postpartum period, following a delivery occurring between 13 March 2020 and recruitment, or cohort 2—pregnant at recruitment, at any trimester as described in Figure 1. In Canada, this period captures the three first waves of the COVID-19 pandemic. During this period, Canada went through a summer (2020) of reopening at the end of the first wave (1st wave, June to August 2020), followed by a 2nd wave in the fall (2020) and a lockdown through winter (2020–2021) (2nd wave, September 2020 to March 2021), followed by a 3rd wave in the spring (2021) as vaccination started rolling out (3rd wave, April to August 2021). The study obtained individual consent and collected data online using SurveyMonkey^®^, a secure platform. Recruitment of participants used diverse methodologies: (i) through press releases and interviews with mainstream media, (ii) online through social media (Facebook, Instagram, Twitter, LinkedIn, and TikTok), (iii) in-person across Canada in community associations for new immigrants, allowing for the recruitment of persons from lower socioeconomic status, (iv) quick response codes were displayed on posters in OB/GYN clinics for persons to directly access the questionnaire with their mobile device. Through recruitment efforts, we aimed to enroll a representative cohort of Canadian pregnant persons or persons who had delivered during the pandemic compared to currently published studies [5,15,16,18,19]. Questionnaires are available in French, English, Mandarin, Spanish, and Portuguese. Team members promoted the study on social media platforms via videos, influencer recruitment with a substantial following (>20,000), mother/child and pregnancy support groups, outpatient and community clinics, hashtag strategies, and through university-affiliated communication specialists. Social media combined with in-person recruitment in crises are appropriate methodologies given the rapidity with which we need answers to pressing questions [20]. Social media recruitment with anonymized data has been used in other similar studies [5,15,16,18,19]. Study eligibility, consent, and baseline data collection were completed electronically; the information was thereafter downloaded on a secure server at CHU Sainte-Justine, Montreal, QC (see https://www.surveymonkey.com/r/etudeCONCEPTIONstudyBL (accessed on 22 November 2021)). All collected data are centralized at CHU Ste-Justine in Montreal, QC, Canada. 

Canadian pregnant persons or persons having delivered after 13 March 2020, aged 18 years or older, and able to read French, English, Spanish, Portuguese or Mandarin were eligible. 

### 2.2. Data Collection

Online questionnaires were pretested and took about 25 min to complete. Variables were self-reported and used standardized validated instruments [21,22,23,24]. We collected the following data in a single questionnaire recruitment as described in Figure 1: (A) Maternal characteristics and physical health (history and since the beginning of pregnancy or referring to the entire pregnancy for those who had delivered): (1) sociodemographic characteristics: gestational age, height, and weight pre-pregnancy to calculate body mass index, maternal age, ethnicity, education, annual income, living arrangements (living alone, yes/no), area of residence (urban, suburban, rural); (2) pregnancy history; (3) health behaviors (e.g., smoking, alcohol use, cannabis use, exercise); (4) comorbidities identified with diagnoses and prescribed medication use (including history and comorbid psychiatric illnesses and psychotropic use), and; (5) work status and employment status changes following the start of the pandemic. (B) Perinatal follow-up and birth plan changes: (1) number of and changes in prenatal visits; (2) partners excluded from delivery rooms; (3) visitors not allowed in hospital; (4) newborn allowed to be with mother after delivery; (5) mother able to visit newborn in the neonatal intensive care unit (if applicable), and; (6) concerns regarding healthcare access as well as maternal and/or newborn COVID-19 diagnosis at delivery (measured on a 4-category ordinal scale with responses ranging from not concerned at all to very concerned). Changes measured were from the result of public health or clinic/hospital recommendations. (C) Mental health: (1) maternal depression, measured using the validated Edinburgh Perinatal Depression Scale (EPDS) in reference to the 7 days before questionnaire completion [21]; (2) anxiety, measured using the validated generalized anxiety disorders (GAD-7) scale in reference to the two weeks before questionnaire completion [22], and; (3) overall stress levels (1:no stress–10:extreme stress scale) since the beginning of COVID-19, measured using the validated Hybrid COVID Stress Scale which includes items from the COVID-19 Impact Scale [23], and from the Coronavirus Perinatal Experiences Impact Survey (COPE-IS) [24]. When persons scored >13 on the EPDS or mentioned thoughts of self-harm (Q10 on the EPDS), they were directed towards support groups. No clinical advice was given to participants.

Depression and anxiety symptoms were measured as continuous variables using the EPDS and GAD-7 instruments, respectively. Depression symptoms were also categorized as moderate to severe (EPDS > 9) and severe (EPDS > 13) [21]. Anxiety symptoms were further defined as moderate to severe (GAD-7 > 9) and severe (GAD-7 > 15) [22]. 

Social fragility was indirectly measured with the sociodemographic variables, and perception/concern measurements with regards to the pandemic described above.

### 2.3. Data Analyses 

Analyses were stratified by pregnancy status at the time of entry in the study (cohort 1 vs. cohort 2, as described in Figure 1). Among those currently pregnant, analyses were stratified on the trimester of pregnancy at recruitment (1st, 2nd, and 3rd). We compared mean maternal depression and anxiety as well as the frequency of severe depression and anxiety, moderate to severe depression and anxiety between study cohorts (cohorts 1 vs. 2), and trimester of pregnancy (cohort 2). Mean stress scores were also compared. For all variables, comparisons using means with standard deviations (SD) or proportions with ranges were done, depending on whether the variables were continuous or categorical, using Student’s t-tests or chi-square statistics, respectively. Missing data for each variable studied are presented; given the study design and recruitment, missing data were not considered in the analyses. 

We quantified determinants of severe maternal depression (EDPS > 13) with univariate and multivariate logistic regression models, accounting for maternal anxiety and stress, socio-demographic and lifestyle variables, gestational age and calendar time at recruitment, and other potential determinants listed above. Estimates (Odds Ratios (OR), and 95% confidence intervals (95%CI)) were calculated. 

Lastly, we looked at maternal depression, anxiety, and stress according to the pandemic period/wave at recruitment (1st wave, 2nd wave, and 3rd wave). 

All statistical analyses were performed using SAS (Version 9.02). 

This study was approved by the CHU Sainte-Justine’s Research Ethics Committee on 23 June 2020 (no. MP-21-2021-2973). 

## 3. Results 

From 23 June 2020 to 17 August 2021, 5885 visited the study online survey, 4445 gave informed consent, and 3200 (71.99%) participated by filling at least one question from the questionnaire (Figure 2). 

### 3.1. Description of the Participants

Participants who gave birth after 13 March 2020, comprising cohort 1, delivered at 38.9 ± 1.9 weeks gestation; pregnant persons at recruitment, comprising cohort 2, were at 24.5 ± 9.7 weeks gestation at recruitment (Table 1). Pregnant persons were more likely to be followed by a family physician and/or midwife as opposed to those who had given birth who were more likely followed by obstetricians (Table 1). 

### 3.2. Maternal Depression 

Participants who had delivered had a significantly higher mean depressive symptoms score compared to those pregnant at recruitment (EPDS, 9.1 (SD, 5.7) vs. 8.4 (SD, 5.3), *p* = 0.009) (Figure 3). Furthermore, pregnant persons who were recruited in their second or third trimester had a significantly higher mean depressive symptoms score compared to those who were recruited in their first trimester (EPDS, 1st trimester, 7.8 (SD, 5.3); 2nd trimester, 8.2 (SD, 5.4); 3rd trimester, 8.7 (SD, 5.2), *p* = 0.007) (Figure 3). Participants who had already delivered were more likely (27.2%) to have *severe* depressive symptoms (EPDS ≥ 13) compared with those who were pregnant (23.4%) (*p* = 0.06) (Figure 4a). Those recruited later in pregnancy (3rd trimester, 25.9%) were more likely to have *severe* depressive symptoms (EPDS ≥ 13) compared with those recruited earlier (1st trimester, 20.7%; 2nd trimester, 22.1%) (*p* = 0.054) (Figure 4a). Similarly, those recruited later in pregnancy (3rd trimester, 43.4%) were significantly more likely to have *moderate to severe* depressive symptoms (EPDS ˃ 9) compared with those recruited earlier (1st trimester, 35.0%; 2nd trimester, 37.8%) (*p* = 0.006) (Figure 4b).

### 3.3. Maternal Anxiety and Overall Stress

Following the same trends observed for depression, the anxiety score (GAD-7) among those who had delivered was significantly higher on average (4.9 (SD, 4.5)) than for those pregnant at recruitment (4.4 (SD, 3.9) (*p* = 0.014) (Figure 5). Although non-significant, *severe* maternal anxiety (GAD-7 ˃ 15) tended to be more prevalent among participants who had delivered (3.5%) than for those who were pregnant (2.4%) (*p* > 0.05) (Figure 6a); the same was observed for *moderate to severe* anxiety (GAD-7 ˃ 10) (12.3% for those who had delivered vs. 9.3% for those who were pregnant, *p* > 0.05) (Figure 6b). Maternal stress measured using a 10-point scale was significantly higher among those who had delivered (5.2 (SD, 2.2)) compared to those who were pregnant at recruitment (4.3 (SD, 2.1) (*p* < 0.001) (Figure 7). Additionally, stress increased as the pregnancy progressed among those pregnant at recruitment (1st trimester, 4.4 (SD, 2.1); 2nd trimester, 4.5 (SD, 2.1); 3rd trimester, 4.7 (SD, 2.1), *p* < 0.001)) (Figure 7).

### 3.4. Predictors of Severe Maternal Depression during Pregnancy 

In multivariate analyses, maternal anxiety (aOR 1.51, 95%CI 1.44–1.59) and maternal stress (aOR 1.35, 95%CI 1.24–1.48) significantly increased the risk of *severe* maternal depression (Table 2). In addition, living in a rural area compared to urban living (aOR 1.64, 95%CI 1.08–2.49), and increasing calendar month of recruitment (proxy for pandemic period/wave) (aOR 1.04, 95%CI, 1.01–1.07) were significant predictors of *severe* maternal depression; increasing household income progressively decreased the risk of severe maternal depression (income ˃ 180,000CAD, aOR 0.26, 95%CI 0.10–0.66) (Table 2). Depression, anxiety, and stress during pregnancy, stratified on COVID-19 periods/waves.

We observed an increase in mean scores for maternal depression (Figure 8), anxiety (Figure 9), and stress (Figure 10) during the 2nd pandemic wave (September 2020 to March 2021) compared to the first wave, as closures were reinstated and the vaccines were not yet available (*p* < 0.001). A decrease in all mean scores was observed during the 3rd wave. 

## 4. Discussion 

In this large-scale Canadian epidemiological study on the maternal mental health impact of the COVID-19 pandemic, we have shown that those who gave birth during the pandemic were more affected mentally than those who were pregnant. In addition, trimester of pregnancy and pandemic period/wave had a different impact on maternal depression, with those in their third trimester and those recruited during the 2nd wave having higher depressive symptoms than the other pregnant participants; suggesting that the closer one came to their planned delivery time, especially during the 2nd wave when closures were reinstated and while the vaccines were not yet available, the higher the level of depression. Thus far, during this pandemic, maternal anxiety, and stress as well as living in rural areas were all significant predictors of severe maternal depression during pregnancy. In this study, we did not include prior history of psychiatric illnesses and psychotropic medication. We assumed a high correlation between these variables and maternal depression, anxiety, and stress at recruitment. 

We have reported higher mean depressive symptom scores in the post-partum period (EPDS, 9.1 (SD, 5.7)), and in pregnancy (EPDS, 8.4 (SD, 5.3)) during the pandemic than what has been reported in other crises and in non-pandemic periods. Indeed, using the same instrument as was used by us to measure depression, the mean depressive symptom scores during the 1998 ice storm (EPDS, 5.5 (SD, 2.6)) [11] and in non-pandemic periods (Norway [25]: EPDS, after delivery, 4.3 (SD, 3.6); 1st trimester, 4.9 (SD, 5.4); 2nd/3rd trimesters, 4.8 (SD, 4.3); Canada/US (Bérard et al. [26]): EPDS ranging from 2.9 to 8.2 depending on antidepressant use during pregnancy) were lower than what we observed. On average, our findings are double that of other crises and non-pandemic periods. [11,25,26] This could be explained by the short duration of the ice storm crisis, which lasted around 30 days and was localized (QC/ON), compared with the current pandemic. At recruitment in this study, the unknown impact of the virus on pregnancy and the baby could explain the increased anxiety and stress, and depression as a result of these two parameters. The growing body of evidence shows that pregnant persons are indeed more at risk of severe disease following COVID-19 infection (e.g., intensive care admission) and death, compared with non-pregnant persons of reproductive age [27]. The restrictions and accommodations for delivery over time can also explain higher levels of depression and anxiety among those who gave birth compared to those who were pregnant at recruitment. 

We observed the highest depression, anxiety, and stress scores in the 3rd trimester compared to the 1st trimester, consistent with the literature [28]. This could be explained by hormonal changes through the pregnancy [28,29] and overall anxiety experienced by women who gave birth during a pandemic. This is probably due to anxiety about exposure to the virus in the hospital and also to COVID regulations about husbands in labor rooms. 

Depression, anxiety, and stress were markedly increased in the 2nd wave, which consisted of the longest lockdowns across Canada, and when no vaccines were available. Indeed, it has been reported that the top impact indicator of major depressive disorder besides daily cases of COVID-19 infection is the reduction of human mobility [30]. Interestingly, all parameters of mental health were lower in the 3rd wave, which coincides with the deployment of vaccination campaigns [23,31,32] representing what may be defined as an end in sight. 

We identified several predictors of severe maternal depression, including maternal anxiety and stress. Anxiety as well as stress are both highly correlated with depression and are predictors of depression [33,34]. Rural compared to urban living was identified as a predictor of severe depression during pregnancy, in line with the literature among Caucasian white women and African Americans alike [35]. Lastly, the calendar month of recruitment, used as a proxy for pandemic period/wave, was identified as a predictor for severe depression. Indeed, as the pandemic progressed, we observed increases in depression, anxiety, and stress levels with a marked increase in the 2nd wave, and a decrease in the 3rd wave, which could explain this finding. 

Our study has many strengths. Indeed, it has a large Canadian sample size allowing us to evaluate the impact of the COVID-19 pandemic on persons who delivered and on pregnant persons’ experience using a robust multi-methods recruitment strategy, and standardized and validated instruments. Data intake was done electronically, increasing the speed with which the study was performed, and thus giving real-time results that remain relevant for public health decisions. The continued recruitment further allowed us to assess the impact of different waves and their implications on maternal mental health. 

Limitations include the absence of a denominator given the recruitment methods. We cannot rule out that our participants were more concerned about the impact of COVID-19 on their pregnancy/delivery than the general population. Furthermore, although internet access could have been a deterrent to participation, we recruited pregnant persons in-person in community clinics (lower socioeconomic status). Although this increased recruitment of a more diverse pregnant population, selection bias cannot be ruled out. Nevertheless, the mean age of participants is representative of the general population of pregnant persons and persons of reproductive age [1]. Our study is among the largest studies on the impact of the COVID-19 pandemic on maternal mental health during pregnancy and postpartum. Ceulemans et al. [18] published survey data in pregnant European persons with similar data intake strategies but only using social media recruitment. Nevertheless, much of their sample was from 2 countries, Norway and The Netherlands, which resulted in unbalanced sample sizes between countries. Our questionnaire was longer than in other studies. This allowed us to collect many variables of interest to answer many research questions (including the impact of the pandemic on maternal mental health) and follow women and children over time (Phase II of the study). However, the impact of the questionnaire length on the participation rate is difficult to assess.

To our knowledge, no study has been conducted to quantify the prevalence of mental health during the perinatal period and identify its predictors in this population through the trimesters of pregnancy. Additionally, the stratification by pandemic waves allows us to assess the impact of public health measures on maternal mental health. 

## 5. Conclusions

The COVID-19 pandemic has had a detrimental impact on mental health throughout pregnancy and after delivery. Depression, anxiety, and stress were markedly increased in the 2nd wave of the pandemic, which was, until now, the most stressful for pregnant persons than other previous crises. Maternal depression, anxiety, and stress were highly correlated. Given that gestational depression/anxiety/stress has been associated with preterm birth and childhood cognitive problems, it is essential to continue following women/children, and develop strategies to reduce COVID-19′s longer-term impact.

## Figures and Tables

**Figure 1 ijerph-19-02926-f001:**
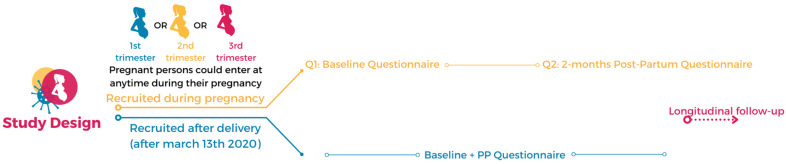
Study design and data collection.

**Figure 2 ijerph-19-02926-f002:**
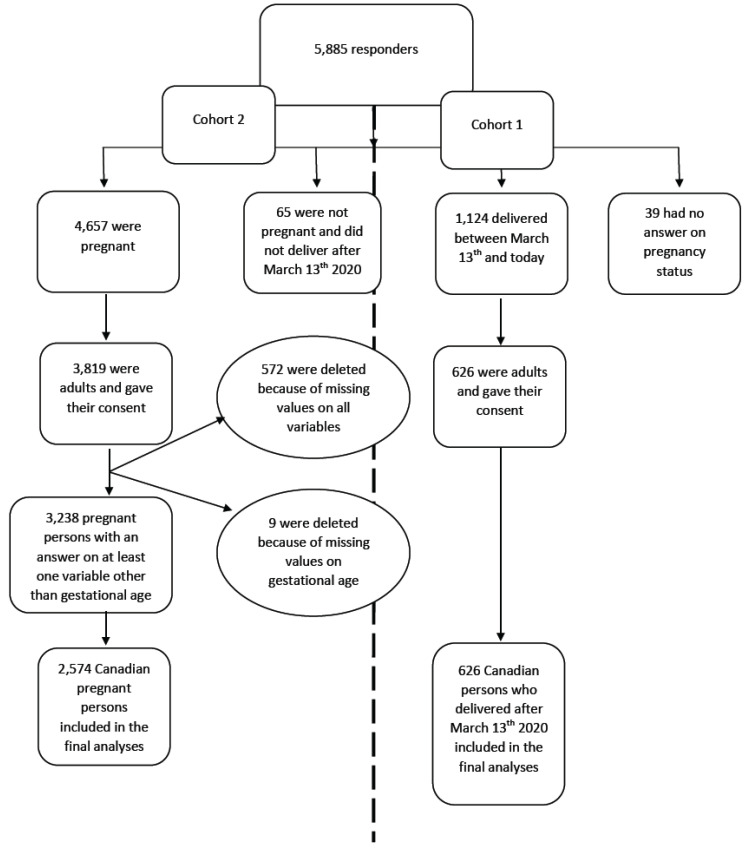
Study flow chart.

**Figure 3 ijerph-19-02926-f003:**
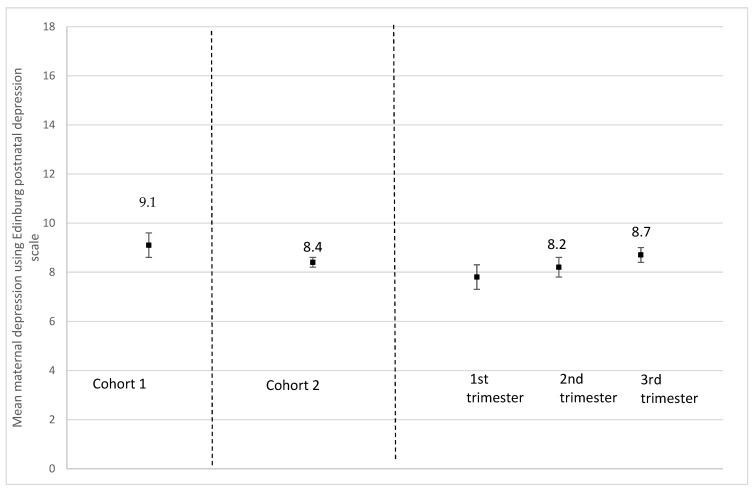
Mean maternal depression scores using the Edinburgh Postnatal Depression Scale (EPDS) according to pregnancy status and trimester of pregnancy. Cohort 1 = women who delivered between 13 March and today at the time of recruitment; *n* = 498; missing values on maternal depression = 128. Cohort 2 = women who were pregnant at the time of the recruitment; *n* = 2251; missing values on maternal depression = 323. *p*-value between cohort 1 and 2 = 0.0091. 1st trimester = women who were at their 1st trimester of pregnancy at the time of recruitment; *n* = 411; missing values on maternal depression = 93. 2nd trimester = women who were at their 2nd trimester of pregnancy at the time of recruitment; *n* = 925; missing values on maternal depression = 107. 3rd trimester = women who were at their 3rd trimester of pregnancy at the time of recruitment; *n* = 915; missing values on maternal depression = 123. *p*-value for all trimesters = 0.0074.

**Figure 4 ijerph-19-02926-f004:**
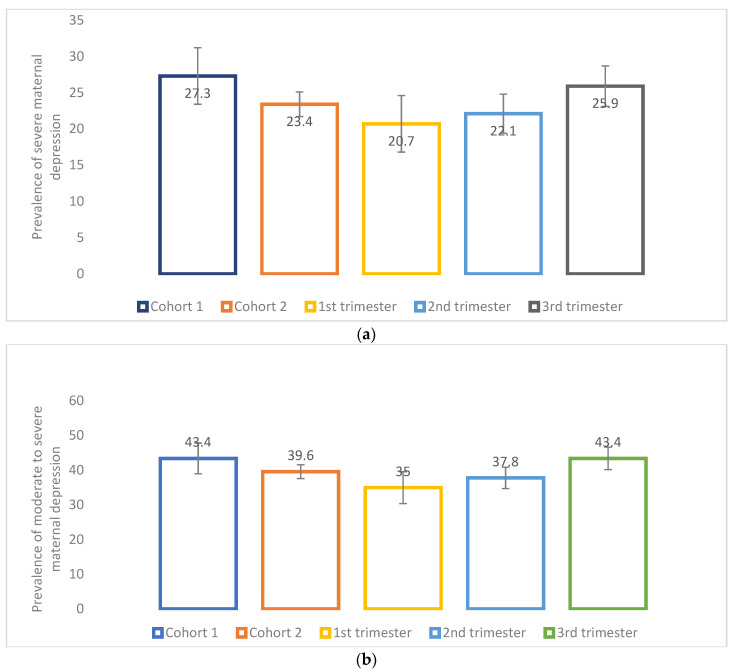
(**a**) Prevalence of severe maternal depressive symptoms using the Edinburgh Postnatal Depression Scale (EPDS) cut-off (≥13) according to pregnancy status and trimester of pregnancy. Cohort 1 = women who delivered between 13 March and today at the time of recruitment; *n* = 498; missing values on maternal depression = 128. Cohort 2 = women who were pregnant at the time of the recruitment; *n* = 2251; missing values on maternal depression = 323. *p*-value between cohort 1 and 2 = 0.0627. 1st trimester = women who were at their 1st trimester of pregnancy at the time of recruitment; *n* = 411; missing values on maternal depression = 93. 2nd trimester = women who were at their 2nd trimester of pregnancy at the time of recruitment; *n* = 925; missing values on maternal depression = 107. 3rd trimester = women who were at their 3rd trimester of pregnancy at the time of recruitment; *n* = 915; missing values on maternal depression = 123. *p*-value for all trimesters = 0.0542. (**b**) Prevalence of moderate to severe maternal depressive symptoms using the Edinburgh Postnatal Depression Scale (EPDS) cut-off (>9) according to pregnancy status and trimester of pregnancy. Cohort 1 = women who delivered between 13 March and today at the time of recruitment; *n* = 498; missing values on maternal depression = 128. Cohort 2 = women who were pregnant at the time of the recruitment; *n* = 2251; missing values on maternal depression = 323. *p*-value between cohort 1 and 2 = 0.1185. 1st trimester = women who were at their 1st trimester of pregnancy at the time of recruitment; *n* = 411; missing values on maternal depression = 93. 2nd trimester = women who were at their 2nd trimester of pregnancy at the time of recruitment; *n* = 925; missing values on maternal depression = 107. 3rd trimester = women who were at their 3rd trimester of pregnancy at the time of recruitment; *n* = 915; missing values on maternal depression = 123. *p*-value for all trimesters = 0.0059 (Compared to the 1st trimester, women in their 3rd trimester had significant results).

**Figure 5 ijerph-19-02926-f005:**
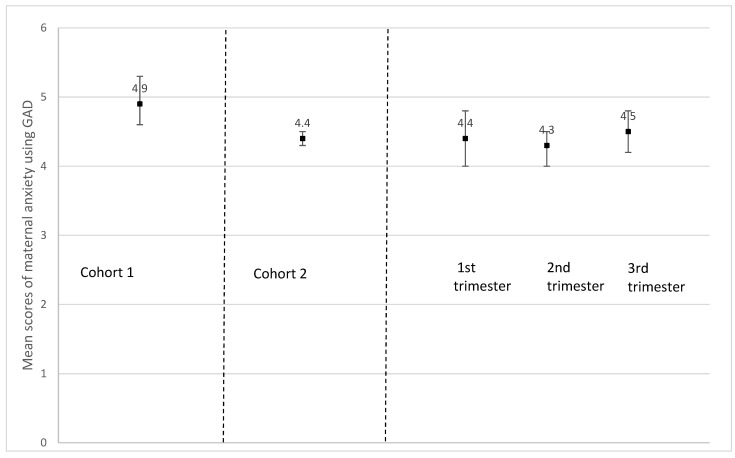
Mean maternal anxiety score using the Generalized Anxiety Disorder 7-item scale (GAD-7) according to pregnancy status and trimester of pregnancy. Cohort 1 = women who delivered between 13 March and today at the time of recruitment; *n* = 489; missing values on maternal depression = 137. Cohort 2 = women who were pregnant at the time of the recruitment; *n* = 2220; missing values on maternal depression = 354. *p*-value between cohort 1 and 2 = 0.0139. 1st trimester = women who were at their 1st trimester of pregnancy at the time of recruitment; *n* = 403; missing values on maternal depression = 101. 2nd trimester = women who were at their 2nd trimester of pregnancy at the time of recruitment; *n* = 917; missing values on maternal depression = 115. 3rd trimester = women who were at their 3rd trimester of pregnancy at the time of recruitment; *n* = 900; missing values on maternal depression = 138. *p*-value for all trimesters = 0.4146.

**Figure 6 ijerph-19-02926-f006:**
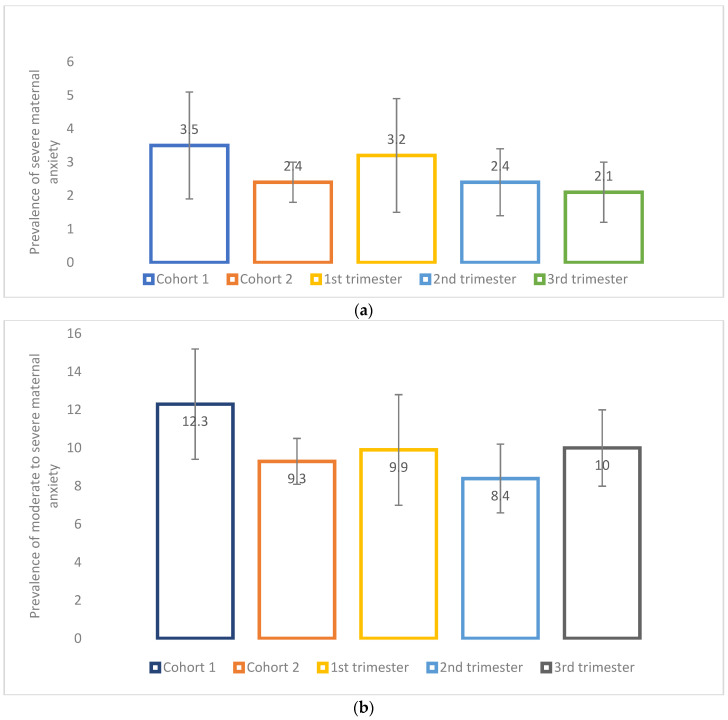
(**a**) Prevalence of severe maternal anxiety symptoms using the Generalized Anxiety Disorder 7-item scale (GAD-7) cut-off (>15) according to pregnancy status and trimester of pregnancy. Cohort 1 = women who delivered between 13 March and today at the time of recruitment; *n* = 489; missing values on maternal depression = 137.Cohort 2 = women who were pregnant at the time of the recruitment; *n* = 2220; missing values on maternal depression = 354. *p*-value between cohort 1 and 2 = 0.1908. 1st trimester = women who were at their 1st trimester of pregnancy at the time of recruitment; *n* = 403; missing values on maternal depression = 101. 2nd trimester = women who were at their 2nd trimester of pregnancy at the time of recruitment; *n* = 917; missing values on maternal depression = 115. 3rd trimester = women who were at their 3rd trimester of pregnancy at the time of recruitment; *n* = 900; missing values on maternal depression = 138. *p*-value for all trimesters = 0.4808 (No significant results when compared 2nd trimester and 3rd trimester to 1st trimester). (**b**) Prevalence of moderate to severe maternal anxiety symptoms using the Generalized Anxiety Disorder 7-item scale (GAD-7) cut-off (>10) according to pregnancy status and trimester of pregnancy. Cohort 1 = women who delivered between 13 March and today at the time of recruitment; *n* = 489; missing values on maternal depression = 137. Cohort 2 = women who were pregnant at the time of the recruitment; *n* = 2220; missing values on maternal depression = 354.np-value between cohort 1 and 2 = 0.0479. 1st trimester = women who were at their 1st trimester of pregnancy at the time of recruitment; *n* = 403; missing values on maternal depression = 101. 2nd trimester = women who were at their 2nd trimester of pregnancy at the time of recruitment; *n* = 917; missing values on maternal depression = 115. 3rd trimester = women who were at their 3rd trimester of pregnancy at the time of recruitment; *n* = 900; missing values on maternal depression = 138. *p*-value for all trimesters = 0.4513 (No significant results when compared 2nd trimester and 3rd trimester to 1st trimester).

**Figure 7 ijerph-19-02926-f007:**
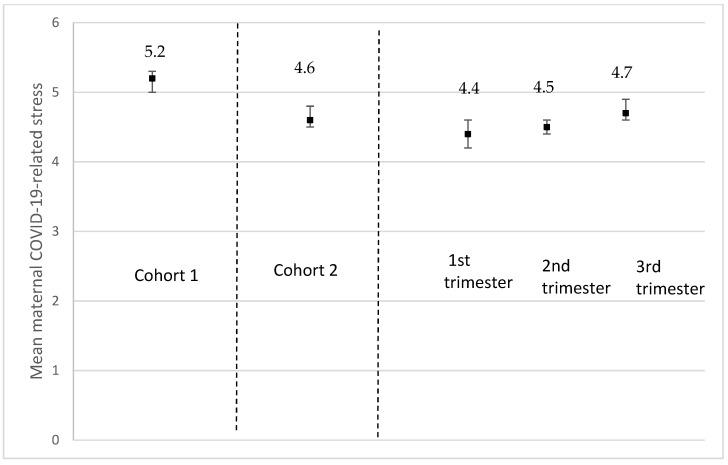
Mean maternal COVID-19-related stress using a scale from 1 (No stress) to 10 (Extreme stress) according to pregnancy status and trimester of pregnancy. Cohort 1 = women who delivered between 13 March and today at the time of recruitment; *n* = 527; missing values on maternal depression = 99. Cohort 2 = women who were pregnant at the time of the recruitment; *n* = 2314; missing values on maternal depression = 260. *p*-value between cohort 1 and 2 <0.0001. 1st trimester = women who were at their 1st trimester of pregnancy at the time of recruitment; *n* = 424; missing values on maternal depression = 80. 2nd trimester = women who were at their 2nd trimester of pregnancy at the time of recruitment; *n* = 948; missing values on maternal depression = 84. 3rd trimester = women who were at their 3rd trimester of pregnancy at the time of recruitment; *n* = 946; missing values on maternal depression = 96. *p*-value for all trimesters <0.0001.

**Figure 8 ijerph-19-02926-f008:**
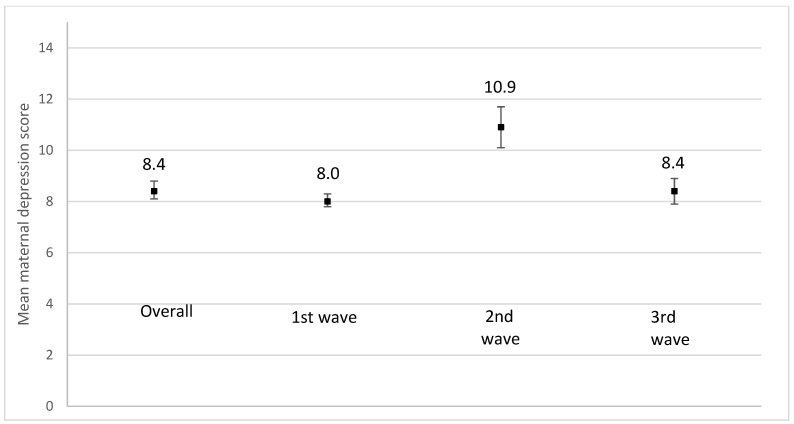
Mean maternal depression scores using the Edinburgh Postnatal Depression Scale (EPDS) according to the recruitment period. 1st wave: recruitment between June and August 2020; *n* = 1605; missing values on maternal depression = 221. 2nd wave: recruitment from September 2020 to March 2021; *n* = 184; missing values on maternal depression = 32. 3rd wave: recruitment from April 2021 to August 2021; *n* = 462; missing values on maternal depression = 70. *p*-value < 0.0001.

**Figure 9 ijerph-19-02926-f009:**
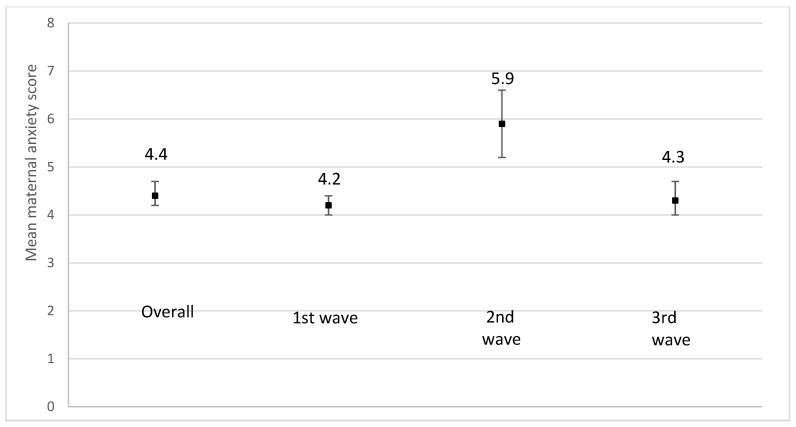
Mean maternal anxiety score using the Generalized Anxiety Disorder 7-item scale (GAD-7) according to the recruitment period. 1st wave: recruitment between June and August 2020; *n* = 1577; missing values on maternal depression = 249. 2nd wave: recruitment from September 2020 to March 2021; *n* = 181; missing values on maternal depression = 35. 3rd wave: recruitment from April 2021 to August 2021; *n* = 462; missing values on maternal depression = 70. *p*-value < 0.0001.

**Figure 10 ijerph-19-02926-f010:**
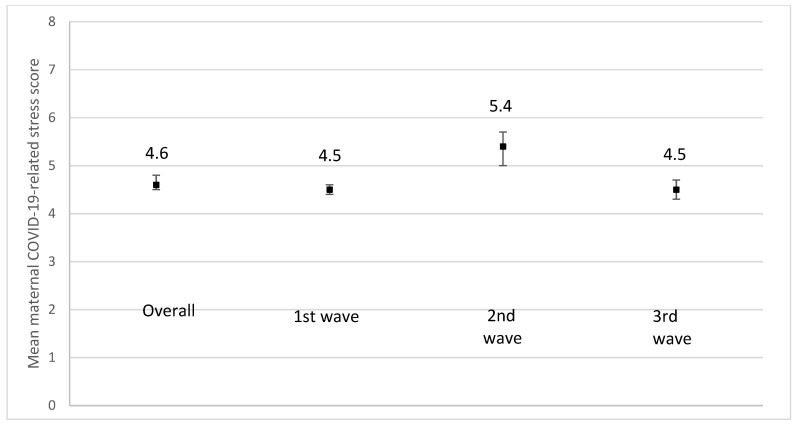
Mean maternal COVID-19-related stress due to the COVID-19 pandemic using a scale from 1 (No stress) to 10 (extreme stress) according to the recruitment period. 1st wave: recruitment between June and August 2020; *n* = 1653; missing values on maternal depression = 173. 2nd wave: recruitment from September 2020 to March 2021; *n* = 188; missing values on maternal depression = 28. 3rd wave: recruitment from April 2021 to August 2021; *n* = 473; missing values on maternal depression = 59. *p*-value < 0.0001.

**Table 1 ijerph-19-02926-t001:** Sociodemographic characteristics of participants according to pregnancy status and trimester of pregnancy.

	Women Who Delivered (Cohort 1) *n* = 626	Women Currently Pregnant (Cohort 2) *n* = 2574	*p*-Value ^+^	First Trimester *n* = 504	Second Trimester *n* = 1032	Third Trimester *n* = 1038	*p*-Value ^/^
Age at recruitment (mean, SD), years	33.2 (4.5)	32.2 (4.3)	<0.0001	32.2 (4.3)	32.3 (4.4)	32.1 (4.2)	0.7320
Missing	4	8		1	2	5	
Gestational age at recruitment (SD), weeks	38.9 (1.9)	24.5 (9.7)	<0.0001	9.8 (3.0)	22.1 (4.0)	34.1 (3.3)	<0.0001
Missing							
Prenatal care follow-up *							
Family physician	201 (32.8)	906 (35.5)	0.2179	196 (39.4)	385 (37.6)	325 (31.6)	0.0023
Obstetrician/Gynaecologist	426 (69.6)	1570 (61.5)	0.0002	275 (55.2)	614 (59.9)	681 (66.1)	<0.0001
Midwife	47 (7.7)	308 (12.1)	0.0020	65 (13.1)	118 (11.5)	125 (12.1)	0.6847
Missing	14	21		6	7	8	
Pre-pregnancy body mass index, kg/m^2^							0.9147
Mean (SD)	25.1 (5.)	25.0 (5.8)	0.6556	24.9 (5.7)	25.0 (5.8)	25.1 (5.7)	
Missing	7	26		2	12	12	
Pregnancy body mass index, kg/m^2^							<0.0001
Mean (SD)	30.1 (5.5)	27.6 (5.9)	<0.0001	25.5 (5.8)	27.0 (5.7)	29.2 (5.8)	
Missing	14	58		13	22	23	
Education, years							0.7740
Mean (SD)	17.4 (3.2)	16.9 (4.5)	0.0042	16.9 (4.7)	17.0 (4.4)	16.9 (4.5)	
Missing	39	98		21	32	45	
Ethnicity-Caucasian/White	540 (90.9)	2242 (90.8)	0.9377	434 (90.8)	901 (90.5)	907 (91.2)	0.8662
Missing	32	105		26	36	43	
Marital status–Living alone	20 (3.4)	52 (2.1)	0.0661	10 (2.1)	19 (1.9)	23 (2.3)	0.8139
Missing	30	86		21	27	38	
Annual household income, CAN$			0.2549				0.3405
<30,000	27 (4.8)	69 (2.9)		4 (0.8)	36 (3.8)	29 (3.1)	
30,001–60,000	49 (8.7)	239 (10.1)		54 (11.5)	91 (9.5)	94 (9.9)	
60,001–90,000	83 (14.7)	374 (15.8)		78 (16.6)	150 (15.7)	146 (15.4)	
90,001–120,000	143 (25.4)	606 (25.5)		126 (26.7)	250 (26.2)	230 (24.3)	
120,001–150,000	101 (17.9)	429 (18.1)		81 (17.2)	175 (18.3)	173 (18.3)	
150,001–180,000	65 (11.5)	299 (12.6)		56 (11.9)	117 (12.3)	126 (13.3)	
>180,000	96 (17.0)	356 (15.0)		72 (15.3)	136 (14.2)	148 (15.7)	
Missing	62	202		33	77	92	
Area of residence			0.2757				0.9796
Urban	268 (45.0)	1065 (42.9)		208 (43.1)	432 (43.2)	425 (42.6)	
Suburban	260 (43.6)	1073 (43.3)		212 (43.9)	431 (43.1)	430 (43.1)	
Rural	68 (11.4)	343 (13.8)		63 (13.0)	138 (13.7)	142 (14.3)	
Missing	30	93		21	31	41	
Recruitment period ^-^							<0.0001
1st COVID wave	-	-	-	341 (67.7)	714 (69.2)	771 (74.3)	
2nd COVID wave	-	-	-	67 (13.3)	95 (9.2)	54 (5.2)	
3rd COVID wave	-	-	-	96 (19.0)	223 (21.6)	213 (20.5)	

These are numbers (column percentages) unless stated otherwise. SD: standard deviation; CAN$: Canadian dollars currency; Cohort 1: women who delivered between 13 March and today at the time of recruitment; Cohort 2: women who were pregnant at the time of recruitment; ^+^ for comparison between cohort 1 and cohort 2; ^/^ for comparison between the three trimesters of pregnancy with 1st trimester as a reference; * women can select multiple options; ^-^ 1st wave: recruitment between June and August 2020, 2nd wave: recruitment from September 2020 to March 2021; 3rd wave: recruitment from April 2021 to August 2021.

**Table 2 ijerph-19-02926-t002:** Determinants of severe maternal depressive symptoms (EPDS > 13) during pregnancy after the start of the COVID-19 pandemic; univariate and multivariate logistic regression models.

Variable	No Depression *n* = 1693 (%)	Severe Depression ^+^ *n* = 511 (%)	Unadjusted OR (95%CI)	Adjusted OR * (95%CI)
Recruitment trimester of pregnancy				
1st trimester	316 (18.7)	80 (15.7)	Reference	Reference
2nd trimester	713 (42.1)	201 (39.3)	1.11 (0.83; 1.49)	1.29 (0.85; 1.95)
3rd trimester	664 (39.2)	230 (45.0)	1.37 (1.03; 1.82)	1.48 (0.97; 2.27)
Maternal anxiety ^/^, mean (SD)	3.1 (2.7)	8.7 (4.3)	1.63 (1.55; 1.71)	**1.51 (1.44; 1.59)**
Maternal stress ^-^, mean (SD)	4.1 (1.9)	6.2 (1.7)	1.86 (1.74; 2.00)	**1.35 (1.24; 1.48)**
Maternal age at recruitment, years	32.3 (4.2)	32.4 (4.2)	1.00 (0.98; 1.03)	1.00 (0.97; 1.04)
Education, years	17.2 (4.3)	16.4 (5.0)	0.96 (0.94; 0.98)	1.00 (0.97; 1.03)
Ethnicity/Race				
Caucasian	1552 (91.7)	464 (90.8)	Reference	Reference
Others	141 (8.3)	47 (9.2)	1.12 (0.79; 1.58)	0.86 (0.51; 1.45)
Annual household income, CAD				
<30,000	33 (1.9)	24 (4.7)	Reference	Reference
30,001–60,000	147 (8.7)	71 (13.9)	0.66 (0.37; 1.21)	0.57 (0.23; 1.37)
60,001–90,000	267 (15.8)	75 (14.7)	0.39 (0.22; 0.69)	**0.38 (0.16; 0.92)**
90,001–120,000	416 (24.6)	138 (27.0)	0.46 (0.26; 0.80)	0.47 (0.20; 1.10)
120,001–150,000	311 (18.4)	89 (17.4)	0.39 (0.22; 0.70)	0.43 (0.18; 1.03)
150,001–180,000	232 (13.7)	61 (11.9)	0.36 (0.20; 0.66)	0.43 (0.17; 1.07)
>180,000	287 (16.9)	53 (10.4)	0.25 (0.14; 0.46)	**0.26 (0.10; 0.66)**
Area of residence				
Urban	735 (43.4)	207 (40.5)	Reference	Reference
Suburban	733 (43.3)	223 (43.6)	1.08 (0.87; 1.34)	0.89 (0.65; 1.22)
Rural	225 (13.3)	81 (15.9)	1.28 (0.95; 1.72)	**1.64 (1.08; 2.49)**
Current number of children				
0	881 (52.0)	232 (45.4)	Reference	Reference
1	583 (34.4)	209 (40.9)	0.74 (0.59; 0.91)	0.76 (0.55; 1.05)
≥2	229 (13.6)	70 (13.7)	0.85 (0.63; 1.16)	0.98 (0.63; 1.53)
Marital status–Living alone				
No	1663 (98.2)	497 (97.3)	Reference	Reference
Yes	30 (1.8)	14 (2.7)	1.56 (0.82; 2.97)	1.77 (0.67; 4.72)
Pre-pregnancy body mass index, mean (SD)	24.9 (5.7)	25.8 (6.0)	1.03 (1.01; 1.04)	1.01 (0.99; 1.04)
Coffee intake				
No	588 (34.7)	162 (31.7)	Reference	Reference
Yes	1105 (65.3)	349 (68.3)	1.15 (0.93; 1.42)	1.24 (0.92; 1.67)
Smoking				
No	1667 (98.5)	495 (96.9)	Reference	Reference
Yes	26 (1.5)	16 (3.1)	2.07 (1.10; 3.89)	0.96 (0.31; 2.94)
Alcohol				
No	1626 (96.0)	491 (96.1)	Reference	Reference
Yes	67 (4.0)	20 (3.9)	0.99 (0.59; 1.65)	0.62 (0.28; 1.39)
Cannabis use				
No	1681 (99.3)	501 (98.0)	Reference	Reference
Yes	12 (0.7)	10 (2.0)	2.80 (1.20; 6.51)	0.75 (0.14; 3.93)
Physical activity				
No change	509 (30.1)	119 (23.3)	Reference	Reference
Start/Increase	298 (17.6)	66 (12.9)	0.95 (0.68; 1.32)	0.99 (0.63; 1.57)
Stop/Decrease	886 (52.3)	326 (63.8)	1.57 (1.24; 1.99)	1.21 (0.87; 1.70)
Multivitamin use during pregnancy				
No	194 (11.5)	67 (13.1)	Reference	Reference
Yes	1499 (88.5)	444 (86.9)	0.86 (0.64; 1.16)	1.03 (0.67; 1.59)
Asthma				
No	1522 (89.9)	445 (87.1)	Reference	Reference
Yes	171 (10.1)	66 (12.9)	1.32 (0.98; 1.79)	0.84 (0.54; 1.32)
Diabetes				
No	1616 (95.5)	475 (92.9)	Reference	Reference
Yes	77 (4.5)	36 (7.1)	1.59 (1.06; 2.40)	0.93 (0.51; 1.72)
Hypertension				
No	1651 (97.5)	484 (94.7)	Reference	Reference
Yes	42 (2.5)	27 (5.3)	2.19 (1.34; 3.59)	1.32 (0.65; 2.68)
Nausea				
No	1351 (79.8)	378 (74.0)	Reference	Reference
Yes	342 (20.2)	133 (26.0)	1.39 (1.10; 1.75)	1.27 (0.92; 1.77)
Thyroid disease				
No	1477 (87.2)	454 (88.8)	Reference	Reference
Yes	216 (12.8)	57 (11.2)	0.86 (0.63; 1.17)	0.64 (0.42; 1.00)
Anemia				
No	1530 (90.4)	437 (85.5)	Reference	Reference
Yes	163 (9.6)	74 (14.5)	1.59 (1.18; 2.14)	1.47 (0.97; 2.23)
Concerns about reduced access to preferred medications				
Not at all concerned	951 (56.2)	195 (38.2)	Reference	Reference
A little concerned	368 (21.7)	131 (25.6)	1.74 (1.35; 2.23)	1.33 (0.93; 1.91)
Moderately/Very concerned	374 (22.1)	185 (36.2)	2.41 (1.91; 3.05)	1.31 (0.92; 1.87)
Concerns about unavailability of primary health care provider for hospital birth				
Not at all concerned	137 (8.1)	24 (4.7)	Reference	Reference
A little concerned	324 (19.1)	61 (11.9)	1.08 (0.64; 1.80)	1.15 (0.56; 2.36)
Moderately/Very concerned	1232 (72.8)	426 (83.4)	1.97 (1.26; 3.09)	0.99 (0.51; 1.94)
Concerns about a possible separation from baby after delivery				
Not at all concerned	238 (14.1)	35 (6.9)	Reference	Reference
A little concerned	326 (19.2)	64 (12.5)	1.34 (0.86; 2.08)	0.62 (0.32; 1.18)
Moderately/Very concerned	1129 (66.7)	412 (80.6)	2.48 (1.71; 3.60)	0.70 (0.36; 1.35)
Concerns about a shorter stay in hospital after delivery				
Not at all concerned	639 (37.7)	131 (25.6)	Reference	Reference
A little concerned	430 (25.4)	102 (20.0)	1.16 (0.87; 1.54)	0.97 (0.64; 1.45)
Moderately/Very concerned	624 (36.9)	278 (54.4)	2.17 (1.72; 2.75)	0.98 (0.67; 1.43)
Concerns about adequate opportunity for skin-to skin contact with newborn				
Not at all concerned	336 (19.9)	47 (9.2)	Reference	Reference
A little concerned	385 (22.7)	93 (18.2)	1.73 (1.18; 2.53)	1.75 (0.94; 3.27)
Moderately/Very concerned	972 (57.4)	371 (72.6)	2.73 (1.97; 3.79)	1.45 (0.75; 2.80)
Concerns about insufficient opportunity to initiate breastfeeding				
Not at all concerned	444 (26.2)	82 (16.1)	Reference	Reference
A little concerned	357 (21.1)	82 (16.1)	1.24 (0.89; 1.74)	0.79 (0.46; 1.35)
Moderately/Very concerned	892 (52.7)	347 (67.8)	2.11 (1.61; 2.75)	0.79 (0.47; 1.33)
Concerns about no visit from family and friends after delivery				
Not at all concerned	356 (21.0)	74 (14.5)	Reference	Reference
A little concerned	348 (20.6)	81 (15.8)	1.12 (0.79; 1.59)	0.88 (0.54; 1.43)
Moderately/Very concerned	989 (58.4)	356 (69.7)	1.73 (1.31; 2.29)	0.95 (0.63; 1.43)
Concerns about reduction of optimal postnatal care				
Not at all concerned	339 (20.0)	38 (7.4)	Reference	Reference
A little concerned	421 (24.9)	77 (15.1)	1.63 (1.08; 2.47)	1.25 (0.71; 2.20)
Moderately/Very concerned	933 (55.1)	396 (77.5)	3.79 (2.65; 5.40)	1.49 (0.86; 2.58)
Concerns about less access to lactation support after discharge from hospital				
Not at all concerned	497 (29.4)	83 (16.2)	Reference	Reference
A little concerned	424 (25.0)	94 (18.4)	1.33 (0.96; 1.83)	1.33 (0.83; 2.12)
Moderately/Very concerned	772 (45.6)	334 (65.4)	2.59 (1.99; 3.38)	1.56 (0.98; 2.46)
Concerns about birth complications due to contracting COVID-19				
Not at all concerned	242 (14.3)	34 (6.7)	Reference	Reference
A little concerned	500 (29.5)	86 (16.8)	1.22 (0.80; 1.87)	0.83 (0.43; 1.62)
Moderately/Very concerned	951 (56.2)	391 (76.5)	2.93 (2.01; 4.27)	0.66 (0.33; 1.29)
Concerns about possible exposure to COVID-19 during pregnancy				
Not at all concerned	121 (7.1)	13 (2.5)	Reference	Reference
A little concerned	469 (27.7)	64 (12.5)	1.27 (0.68; 2.38)	0.59 (0.23; 1.49)
Moderately/Very concerned	1103 (65.2)	434 (85.0)	3.66 (2.04; 6.56)	0.67 (0.26; 1.74)
Concerns about possible exposure to COVID-19 during labour/delivery or shortly thereafter				
Not at all concerned	164 (9.7)	17 (3.3)	Reference	Reference
A little concerned	505 (29.8)	69 (13.5)	1.32 (0.75; 2.31)	0.74 (0.29; 1.87)
Moderately/Very concerned	1024 (60.5)	425 (83.2)	4.00 (2.40; 6.68)	0.93 (0.35; 2.46)
Concerns about baby being infected with COVID-19 after birth				
Not at all concerned	91 (5.4)	6 (1.2)	Reference	Reference
A little concerned	379 (22.4)	62 (12.1)	2.48 (1.04; 5.91)	2.45 (0.67; 8.96)
Moderately/Very concerned	1223 (72.2)	443 (86.7)	5.49 (2.39; 12.64)	1.61 (0.44; 5.98)
Concerns about being infected with COVID-19 and unable to care for newborn				
Not at all concerned	156 (9.2)	14 (2.7)	Reference	Reference
A little concerned	427 (25.2)	61 (11.9)	1.59 (0.87; 2.93)	1.72 (0.65; 4.51)
Moderately/Very concerned	1110 (65.6)	436 (85.3)	4.38 (2.51; 7.65)	2.12 (0.78; 5.79)
Work from home				
No	1036 (61.2)	313 (61.2)	Reference	Reference
Yes	657 (38.8)	198 (38.8)	1.00 (0.81; 1.22)	0.98 (0.73; 1.32)
Loss of job				
No	1580 (93.3)	444 (86.9)	Reference	Reference
Yes	113 (6.7)	67 (13.1)	2.11 (1.53; 2.91)	1.30 (0.80; 2.11)
Decreased take-home pay due to the COVID-19 pandemic				
No	1335 (78.9)	379 (74.2)	Reference	Reference
Yes	358 (21.1)	132 (25.8)	1.30 (1.03; 1.63)	1.15 (0.83; 1.60)
Change in daily routine				
No change	52 (3.1)	12 (2.4)	Reference	Reference
Mild	259 (15.3)	43 (8.4)	0.72 (0.36; 1.46)	0.55 (0.21; 1.43)
Moderate/severe	1382 (81.6)	456 (89.2)	1.43 (0.76; 2.70)	0.57 (0.23; 1.40)
Change in medical health care access				
No change	640 (37.8)	125 (24.4)	Reference	Reference
Mild	855 (50.5)	262 (51.3)	1.57 (1.24; 1.99)	1.12 (0.81; 1.54)
Moderate/severe	198 (11.7)	124 (24.3)	3.21 (2.39; 4.31)	1.32 (0.87; 2.01)
Change in access to family, extended family and non-family social support				
No change	71 (4.2)	14 (2.7)	Reference	Reference
Mild	781 (46.1)	119 (23.3)	0.77 (0.42; 1.42)	1.13 (0.45; 2.82)
Moderate/severe	841 (49.7)	378 (74.0)	2.28 (1.27; 4.10)	1.68 (0.68; 4.14)
Recruitment time (month and year)	-	-	-	**1.04 (1.01; 1.07)**

*n* = 2204 because of missing values on maternal depression, anxiety and stress (*n* = 370); SD: standard deviation; CAN$: Canadian dollars currency; ^+^ severe depression: EDPS ≥ 13; ^/^ Using Generalized Anxiety Disorder; ^-^ Using Overall maternal stress related to COVID-19 scale from 0 (no stress) to 10 (extreme stress). * adjusted for all variables in the table and time. All bold numbers have a significant difference (*p* < 0.05).

## Data Availability

Anonymised individual-level data from the study including data dictionaries, data collection tools will be made available upon request. Requests for access will be reviewed by a data access committee.

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
