# Peer review of "The COVID-19 Pandemic Impacted Maternal Mental Health Differently Depending on Pregnancy Status and Trimester of Gestation"

_ijerph, 2022, doi:10.3390/ijerph19052926_

Round 1

Reviewer 1 Report

The manuscript "The COVID-19 pandemic impacted maternal mental health differently depending on pregnancy status and trimester of gestation" is an extremely interesting contribution evaluating the mental health of women during different pregnancy phases in Canada.

The study is extremely well designed and the manuscript is well written and follows the description standards of STROBE for descriptive epidemiological studies.

Authors also do an excellent job summarizing both the strengths and weakness of the study in the discussion, and, all in all, this study might become a landmark in the field of measuring mental health during a prolonged emergency.

So not many comments, but a few suggestions of things that need to be improved to make the manuscript excellent.

Minor comments

1) Please, re-draw figure 1 increasing font size. Currently fonts are way to small compared to the text in the main text.

2) The flow chart needs better tags. What are the different "NO" options. Please try to make the figure more clear so that it becomes intuitive to readers that don t carefully read the methods. For example, make clear what are the 1st and 2nd cohorts and so on.

3) In figure 4 please include the 95% CI for the prevalence estimates. That information is  currently not presented and showing that they are likely small (given the large sample size) would emphasize the design qualities of the study.

4) Please consider also including the 95% CI for the prevalence estimates presented in the two panels of figure 6. That information is  currently not presented and showing that they are likely small (given the large sample size) would emphasize the design qualities of the study.

5) The ethical clearance is missing details such as data, number etc (2nd line in page 5 of 20).

Author Response

Reviewer 1

The manuscript "The COVID-19 pandemic impacted maternal mental health differently depending on pregnancy status and trimester of gestation" is an extremely interesting contribution evaluating the mental health of women during different pregnancy phases in Canada.

The study is extremely well designed and the manuscript is well written and follows the description standards of STROBE for descriptive epidemiological studies.

Authors also do an excellent job summarizing both the strengths and weakness of the study in the discussion, and, all in all, this study might become a landmark in the field of measuring mental health during a prolonged emergency.

So not many comments, but a few suggestions of things that need to be improved to make the manuscript excellent.

Minor comments

1) Please, re-draw figure 1 increasing font size. Currently fonts are way to small compared to the text in the main text.

Response: We thank the reviewer for this comment. We have revised Figure 1 in the revised version of this manuscript as suggested.

2) The flow chart needs better tags. What are the different "NO" options. Please try to make the figure more clear so that it becomes intuitive to readers that don t carefully read the methods. For example, make clear what are the 1st and 2nd cohorts and so on.

Response: We thank the reviewer for this comment. We have revised Figure 2 as suggested and made it clear in all other figures what Cohort 1 and 2 refer to.

3) In figure 4 please include the 95% CI for the prevalence estimates. That information is currently not presented and showing that they are likely small (given the large sample size) would emphasize the design qualities of the study.

Response: We thank the reviewer for this comment. In this revised version, we included the 95% CI in Figures 4a and 4b as suggested. 

4) Please consider also including the 95% CI for the prevalence estimates presented in the two panels of figure 6. That information is currently not presented and showing that they are likely small (given the large sample size) would emphasize the design qualities of the study.

Response: We thank the reviewer for this comment. In this revised version, we included the 95% CI in Figures 6a and 6b as suggested. 

5) The ethical clearance is missing details such as data, number etc (2nd line in page 5 of 20).

Response: We thank the reviewer for pointing this out. P. 5, last paragraph has been revised as:

‘This study was approved by the CHU Sainte-Justine's Research Ethics Committee on June 23rd, 2020 (no. MP-21-2021-2973).’ 

Reviewer 2 Report

Thank you to let me review this interesting manuscript. I have some observations about this manuscript: 

Introduction: The prevalence of depression, anxiety and stress in pregnant women is not discussed except in relation to the pandemic and exposure to crises (climatic events). 

Obstetric complications resulting from stress and depression in pregnancy are also mentioned, such as low birth weight, prematurity and postpartum depression without specifying the incidence of these and their connection with other maternal pathologies that are not mentioned (use of drugs, alcohol, smoking in pregnancy, pre-existing psychiatric pathologies, use of psychotropic drugs, poor controls in pregnancy, obstetric history). 

Materials and methods: Robust and well organized recruitment strategy (standardized rating scales). By using a computerized method to recruit patients and receive questionnaires they were able to reduce technical time and to recruit many patients from various social categories, making the sample under review representative of the general population.

Very long questionnaire (25 minutes), high number of patients eliminated due to lack of answers to all questions (572).

Apparently, the questions in the questionnaire do not investigate maternal psychiatric illness and the use of psychotropic drugs, the use of drugs (except cannabis), the history of social fragility.

Discussion: The target population is also compared to populations of pregnant and postpartum women outside the pandemic analyzed in other studies (snow storm 1998) with the same rating scales (EPDS). With what certainty do we know that the two studies can be compared and that the two populations are also comparable? Also calculating the time difference between the two studies (1998 vs 2020) and that the storm in question lasted 30 days and was localized (Quebec / Ontario). Again, no previous psychiatric disorders or other confounding factors are mentioned, which are often decisive.

Author Response

Reviewer 2

Thank you to let me review this interesting manuscript. I have some observations about this manuscript: 

Introduction: The prevalence of depression, anxiety and stress in pregnant women is not discussed except in relation to the pandemic and exposure to crises (climatic events). 

Obstetric complications resulting from stress and depression in pregnancy are also mentioned, such as low birth weight, prematurity and postpartum depression without specifying the incidence of these and their connection with other maternal pathologies that are not mentioned (use of drugs, alcohol, smoking in pregnancy, pre-existing psychiatric pathologies, use of psychotropic drugs, poor controls in pregnancy, obstetric history). 

Response: The CONCEPTION study is on-going and will be for a long time – we plan to follow women and children for at least 10 years. Hence, we have collected data on many variables at different points in time. Although it is true that we have data on pregnancy outcomes among participants in cohort 1 (those who had delivered at recruitment), we do not for those in cohort 2 (these participants are still pregnant). Hence, the current manuscript aims to quantify differences in maternal mental health according to pregnancy/delivery status, and trimester of recruitment. We are planning to study the impact of maternal mental health measures on pregnancy outcomes when these will be available on all of our CONCEPTION participants.

Materials and methods: Robust and well organized recruitment strategy (standardized rating scales). By using a computerized method to recruit patients and receive questionnaires they were able to reduce technical time and to recruit many patients from various social categories, making the sample under review representative of the general population.

Response: We thank the reviewer for this comment/observation.

Very long questionnaire (25 minutes), high number of patients eliminated due to lack of answers to all questions (572).

Response: We thank the reviewer for this comment. Indeed, our questionnaire was longer than in other studies. This allowed us to collect many variables of interest to answer many research questions (including the impact of the pandemic on maternal mental health) and follow women and children over time (Phase II of the study). Although this likely had an impact on the participation rate, we suspect that many responders were curious about the study, accessed the social media pages but never intended to participate for various reasons (not eligible, not interest OR the questionnaire was too long). This is difficult to quantify. Nevertheless we added the following sentence in the Discussion section (paragraph on limitations), p. 19, end of 2nd paragraph:

‘Our questionnaire was longer than in other studies. This allowed us to collect many variables of interest to answer many research questions (including the impact of the pandemic on maternal mental health) and follow women and children over time (Phase II of the study). Although this likely had an impact on the participation rate, we suspect that many responders were curious about the study, accessed the social media pages but never intended to participate for various reasons (not eligible, not interest or the questionnaire was too long). Thus, the impact of the questionnaire length on participation rate is difficult to assess.’

Apparently, the questions in the questionnaire do not investigate maternal psychiatric illness and the use of psychotropic drugs, the use of drugs (except cannabis), the history of social fragility.

Response: History and comorbid psychiatric illnesses and psychotropic use are collected and self-reported by participants. In addition, maternal depression (EPDS), anxiety (GAD-7) and stress (visual analogue scale) are collected at the time of recruitment in both cohorts, using validated scales. Social fragility is indirectly measured with our sociodemographic variables, and perception/concern measurements with regards to the pandemic. This was described in the Methods section, but an enhanced description taking the reviewer’s comments into account has been included in the revised version of this manuscript on p.4.

The beginning of the first paragraph of p. 4 now reads as:

‘….pregnancy history; 4) health behaviors (e.g. smoking, alcohol use, cannabis use, exercise); 5) comorbidities identified with diagnoses and prescribed medication use (including history and comorbid psychiatric illnesses and psychotropic use); and 6) work status and employment status changes following the start of the pandemic. B) Perinatal follow-up…’

We have also added the following sentence on p. 4, 3rd paragraph (above Figure 1):

‘Social fragility was indirectly measured with the sociodemographic variables, and perception/concern measurements with regards to the pandemic, described above.’

Discussion: The target population is also compared to populations of pregnant and postpartum women outside the pandemic analyzed in other studies (snow storm 1998) with the same rating scales (EPDS). With what certainty do we know that the two studies can be compared and that the two populations are also comparable? Also calculating the time difference between the two studies (1998 vs 2020) and that the storm in question lasted 30 days and was localized (Quebec / Ontario). Again, no previous psychiatric disorders or other confounding factors are mentioned, which are often decisive.

Response: We agree with the reviewer that the current pandemic and the 1998 winter storm are not similar in nature; this was discussed in the Discussion section, p. 18, 2nd paragraph. Other comparisons were also made with findings from other studies performed outside of the pandemic (in fact the Winter storm estimates are similar to estimates in non-pandemic periods, which is not the same for the COVID-19 pandemic findings presented in our manuscript). However, this was done mainly to emphasize the impact of the COVID-19 pandemic on maternal mental health.

Here is the Discussion section, p. 18, 2nd paragraph:

‘We have reported higher mean depressive symptom scores in the post-partum period (EPDS, 9.1 (SD, 5.7)), and in pregnancy (EPDS, 8.4 (SD, 5.3)) during the pandemic than what has been reported in other crises and in non-pandemic periods. Indeed, using the same instrument as was used by us to measure depression, the mean depressive symptom scores during the 1998 ice storm (EPDS, 5.5 (SD, 2.6))(11) and in non-pandemic periods (Norway(25): EPDS, after delivery, 4.3 (SD, 3.6); 1st trimester, 4.9 (SD, 5.4); 2nd/3rd trimesters, 4.8 (SD, 4.3); Canada/US (Bérard et al. (26)): EPDS ranging from 2.9 to 8.2 depending on antidepressant use during pregnancy) were lower than what we observed. On average, our findings are double that of other crises and non-pandemic periods. (11, 25, 26) This could be explained by the short duration of the ice storm crisis, which lasted around 30 days and was localized (Quebec/Ontario), compared with the current pandemic. At recruitment in this study, the unknown impact of the virus on pregnancy and the baby could explain the increased anxiety and stress, and depression as a result of these two parameters. The growing body of evidence shows that pregnant persons are indeed more at risk of severe disease following COVID-19 infection (e.g., intensive care admission) and death, compared with non-pregnant persons of reproductive age.(27) The restrictions and accommodations for delivery over time can also explain higher levels of depression and anxiety among those who gave birth compared to those who were pregnant at recruitment.’

In the multivariate models, we have taken history of psychiatric disorders and use of medications as well as other determinants into account in order to explain de variations in severe maternal depression status (Table 2, p. 13). Although we have included many maternal comorbid conditions in the model, we have not included history of psychiatric illnesses and psychotropic medication use because these were too highly correlated with our measures of maternal anxiety (GAD-7), and stress in relation with our dependent variable, severe maternal depression (EDPS). We have found that, maternal anxiety and stress are the 2 major determinants of maternal depression during pregnancy (p. 13, Table 2). In order to clarify this, we have added the following sentence in our revised manuscript (Discussion section, p. 18, end of 1stparagraph).

‘Although we considered many maternal comorbid conditions in our analyses, we have not included history of psychiatric illnesses and psychotropic medication use because these were too highly correlated with maternal depression, anxiety, and stress at recruitment.’